# High Performance Tunable Catalysts Prepared by Using 3D Printing

**DOI:** 10.3390/ma14175017

**Published:** 2021-09-02

**Authors:** Cristian Yesid Chaparro-Garnica, Esther Bailón-García, Arantxa Davó-Quiñonero, Patrick Da Costa, Dolores Lozano-Castelló, Agustín Bueno-López

**Affiliations:** 1Department of Inorganic Chemistry, University of Alicante, Carretera de San Vicente del Raspeig s/n, 03080 Alicante, Spain; cristian.chaparro@gcloud.ua.es (C.Y.C.-G.); arantxa.davo@tcd.ie (A.D.-Q.); d.lozano@ua.es (D.L.-C.); agus@ua.es (A.B.-L.); 2Carbon Materials Research Group, Department of Inorganic Chemistry, Faculty of Sciences, Campus Fuentenueva s/n, University of Granada, 18071 Granada, Spain; 3Institut Jean Le Rond d’Alembert, CNRS UMR 7190, 2 Place de la Gare de Ceinture, Sorbonne Université, 78210 Saint Cyr L’Ecole, France; patrick.da_costa@sorbonne-universite.fr

**Keywords:** 3D-printing, carbon monoliths, morphology control, porosity control, CO_2_ methanation

## Abstract

Honeycomb monoliths are the preferred supports in many industrial heterogeneous catalysis reactions, but current extrusion synthesis only allows obtaining parallel channels. Here, we demonstrate that 3D printing opens new design possibilities that outperform conventional catalysts. High performance carbon integral monoliths have been prepared with a complex network of interconnected channels and have been tested for carbon dioxide hydrogenation to methane after loading a Ni/CeO_2_ active phase. CO_2_ methanation rate is enhanced by 25% at 300 °C because the novel design forces turbulent flow into the channels network. The methodology and monoliths developed can be applied to other heterogeneous catalysis reactions, and open new synthesis options based on 3D printing to manufacture tailored heterogeneous catalysts.

## 1. Introduction

Heterogeneous catalysis is involved in countless industrial processes, and honeycomb monoliths are widely used because of their benefits with regard to packed beds, like lower energy input requirements due to the lower pressure drop, easy replacement, and higher mechanical stability. However, the geometry of honeycomb monoliths is limited to designs obtained by extrusion, consisting of parallel channels that impose laminar flow regime to the fluid [1]. This limits the efficiency of the usually expensive active phases loaded on the channel walls, as laminar flow hinders radial diffusion of the fluid, and heat and mass radial diffusion are impeded.

The use of the 3D printing technology to prepare monoliths with specific configurations would be a breakthrough in the field of catalysis, in line with the successful advances done in medicine or materials science among other areas. Three-dimensional printing is a very powerful approach for fabrication of prototypes, being of special interest for research purposes, as demonstrated in this article. 

Several approaches have been developed to obtain 3D-printed monoliths in recent years. The easiest and most extended 3D-printing technology is the Fused Deposition Modeling (FDM), where a polymeric filament is melted in a heated nozzle and is deposited with the desired design. Research efforts have been focused on the incorporation of the active component, including metals, metal oxides, or advanced carbon materials [2,3,4,5,6,7,8] (e.g., graphene) into the polymer filaments. However, obtaining this filament is not easy since the amount of active phase incorporated to the filament is limited by the FDM printing conditions and part of the expensive active phase is wasted as it is embedded into the polymer bulk [9]. This is one of the main handicaps of this approach, since overloading of generally expensive active phases is required to obtain proper conversion levels. Moreover, the poor thermal and chemical stabilities of the polymeric matrix limit their application in catalysis. 

Another explored approach is to prepare an injectable active phase-paste which is used to print the desired piece by extrusion of the paste [10,11,12,13,14,15]. This direct ink writing consists in the extrusion of concentrated suspensions formulated of main material together with additives to get appropriate viscoelastic behavior, which is the main requirement of this technology [16]. Then, the process requires a subsequent sintering process at controlled temperature of the printed material to obtain a solid ceramic part and achieve the final strength, in which organic additives can be removed. Using this approach as an example, 3D printed catalytic converters have been obtained with non-linear channel structures and tested in the oxidation of methane [17,18]. The design consisted of simple repeating angular offset between adjacent layers, which creates turbulence inside the channels. Higher methane conversion was obtained using structures with more turbulence, which clearly manifest the benefits that the 3D-printing could entails in catalytic applications. Nonetheless, it is not possible to obtain highly dense pieces, which limits their application in the industry and the use of a slurry that contains various additives, such as plasticizers, dispersants, surfactants, binders, defoamers, lubricants, etc. They contribute to the formation of defects during sintering related to the evaporation of the aforementioned additives, which lead to volumetric shrinkage and crack formation, which considerably reduce the mechanical properties of the ceramics designs [16].

Other 3D printing technologies are rising aiming to obtain ceramic designs with complex geometries such as stereolithography (SL), digital light processing (DLP) and two-photon polymerization (TPP) [10,11,12,13,19]. These slurry-based ceramic 3D printing technologies make possible to photo-polymerizate pre-ceramic polymers and thus, to obtain ceramic monolithic materials with complex geometries. Around 50–70 of ceramic phase has been successfully introduced into 3D printed materials [19]. However, the presence of polymeric matrix still limits the application of these 3D printed materials in catalytic applications. In order to avoid the limitations of polymeric matrix, powder-based 3D printing technologies are being explored to obtain ceramics or metals 3D printed materials, such as selective laser sintering (SLS) or selective laser melting (SLM). However, this technology is much more expensive, and larger and more complex devices are required that are not as available as FDM printers, which limits the utilization of the metal and ceramic technologies. Therefore, despite the great progress made in the fabrication of 3D printed ceramics materials, the optimization of processing parameters and post-processing are obstacles for the wider use of 3D printing of ceramics [19], especially in catalysis.

Alternatively, other potential approach would consist in using a polymeric 3D-printed template, to be filled with a paste (i.e., alumina, cordierite or zeolite) dispersed in a solvent media [20] and finally removed by calcination. However, access of this viscous paste to the entire template design is very complicated, as well as the removal of air from the inside of the molds, which creates hollow spaces and fractures during the calcination process. These handicaps limit the design versatility of 3D printing in this case. 

For all these reasons, the potential of 3D-printing technology has not likely been significantly exploited in heterogeneous catalysis yet, and as far as we know, improvements of novel channel designs in catalysis have not been extensively reported to date. As an advanced solution, we report here the synthesis of integral carbon gels-based monoliths with unique properties by combining the versatility of the sol-gel process and the 3D-printing technology. High performance carbon monoliths have been prepared with a complex network of interconnected channels and have been tested for carbon dioxide hydrogenation to methane after loading a Ni/CeO_2_ active phase. Therefore, in this manuscript, DMF printing technology is used as an impressive tool to obtain advanced materials of great interest in many applications. Therefore, we demonstrate that additive manufacturing technologies provide a disruptive transformation in how catalysts are designed and manufactured and provide a new synthesis method in which 3D-printing technologies is incorporated to create new and advanced carbon-based catalysts, which cannot be obtained by the traditional technology.

## 2. Materials and Methods

### 2.1. Active Phase Synthesis

Ni/CeO_2_ was selected as active phase for CO_2_ methanation reaction based on bibliographic results [21,22,23]. Ni/CeO_2_ active phase was synthesized by incipient wetness impregnation of a previously synthesized CeO_2_ support. For that, Ce(NO_3_)_3_·6H_2_O (99.5%, Alfa Aesar) was calcined at 500 °C for 4 h at a heating rate of 5 °C/min. Then, this CeO_2_ support was impregnated with an ethanolic solution of Ni(NO_3_)_2_·6H_2_O (Sigma-Aldrich, Darmstadt, Germany), dried a 50 °C overnight and finally, calcined at 600 °C in static air for 5 h (heating rate of 5 °C/min) to decompose the precursor salt and to obtain the final Ni/CeO_2_ active phase.

### 2.2. 3D-Printed Carbon Monoliths

Integral carbon monoliths with cylindrical shape (1 cm diameter; 1.5 cm length) and specific channel architectures were synthesized by combining the 3D printing technology and the resorcinol-formaldehyde (RF) sol-gel polycondensation. First, the templates were manufactured with two different designs by the 3D printing of a high chemical resistance polymer (polyester, CPE+): conventional honeycomb design with straight channels (CD) and advanced tortuous design (AD) with channels that split and join successively along the monolith length to create a tortuous path. The templates were printed with a channel diameter of 1 mm using a Ultimaker 2+ 3D printer. Subsequently, the template and a mixture of resorcinol (R), formaldehyde (F) and water (W) were deposited inside glass tubes and the tubes were sealed and subjected to a polymerization and aged thermal program consisting of 1 day at room temperature, 1 day at 50 °C and 5 days at 80 °C. The R/F molar ratio was 1/2 and the R/W molar ratio was 1/17. To analyze the influence of the porous structure of the carbon monolith on the active phase dispersion and catalytic performance, monoliths with R/W molar ratios of 1/13 and 1/15 were also prepared for the conventional design. After the curing process, the organic gels were extracted from the glass tubes and introduced in acetone for 3 days (the acetone was changed twice daily) to exchange the water retained in the porous structure by acetone, and thus, prevent the porosity collapse during the drying process. Finally, the organic monoliths were dried and carbonized at 900 °C for 2 h at a heating rate of 1.5 °C/min. In the carbonization step, the polymeric template was removed and carbon monoliths with designed morphology and adapted texture properties obtained.

### 2.3. Active-Phase Loading into the 3D-Printed Carbon Monoliths

The previously synthesized Ni/CeO_2_ active phase was loaded on the carbon surface by dip-coating of the monoliths into a Ni/CeO_2_ ethanolic suspension (15% wt.) under stirring. After immersion, the monoliths were dried at room temperature in horizontal rotation for 2 h and then, at 100 °C in an oven. The excess of active phase that can obstruct the channels and that is not adhered to the support was removed with compressed air. Finally, the monoliths were treated at 500 °C in inert atmosphere for 2 h at a heating rate of 3 °C/min. To determine the amount of Ni/CeO_2_ loaded on each carbon monolith, the weight was monitored before and after the dip coating process. Furthermore, the amounts of active phase anchored in the different monoliths were also confirmed by burning the supported catalysts at 700 °C for 2 h. The amount of active phase incorporated in all monoliths was 115 ± 8 mg (0.17 g active phase/g of monolith).

### 2.4. Catalyst Characterization

The morphology of carbon monoliths and the Ni/CeO_2_ distribution on the monoliths were analyzed by scanning electron microscopy (SEM) in a S-3000N microscope (HITACHI, Tokio, Japan).

Mercury porosimetry was performed using a POREMASTER-60 GT equipment from QUANTACHROME INSTRUMENTS (Sulzemoos, Germany). N_2_ and CO_2_ isotherms were recorded at −196 and 0 °C, respectively, using an Autosorb-6B equipment (Quantachrome instruments, Sulzemoos, Germany). The samples were degassed at 60 °C for 12 h and 110 °C for 8 h before the mercury intrusion/extrusion and gas adsorption analysis, respectively. The micropore volume and micropore width were obtained by applying the Dubinin-Radushkevich (DR) equation to the adsorption data. The micropores surface area (S_mic_) was calculated from the micropore volume assuming slit pores.

The reducibility of the supported catalysts was studied by temperature programmed reduction experiments with H_2_ using a Belcat-M from Bel Japan Inc. (Osaka, Japan). X-ray diffractograms were recorded in a Miniflex II diffractometer (Rigaku, Neu-Isenburg, Alemania) in order to study the crystallinity of the catalysts. The Raman spectra were recorded using a RFS/100 dispersive Raman spectrometer from Bruker (Bremen, Germany) coupled to a microscope with a He-Ne laser source (632.8 nm). 

The CO_2_ methanation experiments were performed in a U-shaped glass reactor using the active phase/3D-printed carbon monoliths as catalysts (weight of 0.760 g). The catalysts were pre-treated to reduce the nickel species at 500 °C for 1 h in a flow of 5% H_2_/Ar (100 mL/min). After cooling to 200 °C in inert gas, the reactor was fed with the reaction mixture, which is composed of 15% CO_2_, 60% H_2_ and Ar balance up to 100 mL/min. Under these conditions, the gas hourly space velocity (GHSV) was 5093 h^−1^. The temperature was increased from 200 to 450 °C with a heating rate of 10 °C/min, in steps of 50 °C and the concentration of reactants and products at the reactor outflow was determined by chromatographic analysis using a Varian GC4900 (Agilent Technologies Spain, S.L, Madrid, España) once the steady state was reached at each temperature step.

CO_2_ conversion (C_CO2_), CH_4_ selectivity (S_CH4_) and reaction rate (r, μmol CO_2_ converted g^−1^ s^−1^) were calculated as:$$C_{CO_{2}}=\frac{F_{CO_{2}}^{in}-F_{CO_{2}}^{out}}{F_{CO_{2}}^{in}}\times 100$$
$$S_{CH_{4}}=\frac{F_{CH_{4}}}{F_{CH_{4}}+F_{CO}}\times 100.$$
$$r=\frac{F_{CO_{2}}^{in}-F_{CO_{2}}^{out}}{W}\times 10^{6}$$
where $F_{CO_{2}}^{in}$ and $F_{CO_{2}}^{out}$ are the CO_2_ flows at the reactor inlet and outlet, respectively. $F_{CH_{4}}$ and $F_{CO}$ are the CH_4_ and CO flows in the outlet gas. W is the weight of catalyst used in the experiment.

## 3. Results and Discussion

The synthesis of integral carbon gels-based monoliths with unique properties by combining the versatility of the sol-gel process and the 3D-printing technology is proposed in this work. The proposed approach (Figure 1) consists of the use of a 3D-printed polymer mold with the desired geometry that is filled with a carbon gel precursor solution. This aqueous solution has rheological properties similar to those of water and is able to access details of the mold in the micrometers range, as shown in SEM images set out below. After filling the molds, the precursor mixture is subjected to a controlled polymerization, gelation and curing process, finally being dried and carbonized. The surface chemistry and the textural properties of carbon can be tuned by controlling the sol-gel polymerization conditions combined with the freedom of design provided by 3D printing, the features of the catalyst support can be tuned according to the application requirements, being a very flexible and straightforward approach. First, the design made by 3D printing controls the flow conditions into the catalyst body. Second, the tunable textural and chemical properties of carbon can be set to favor the dispersion and anchoring of the active phases, as well as the adsorption and interaction of the reagents with such active phases, promoting an efficient use of resources and raw materials. This section may be divided by subheadings. It should provide a concise and precise description of the experimental results, their interpretation, as well as the experimental conclusions that can be drawn.

At this way, integral carbon monoliths with cylindrical shape (1 cm diameter; 1.5 cm length) were synthesized with two different channel geometries: conventional honeycombs with straight channels (Figure 2A) and advanced tortuous channels, (Figure 2B). These 3D-printed templates were filled with the carbon gel precursor mixture: a mixture formed by resorcinol, formaldehyde and water. The resorcinol/water (R/W) ratio was ranged from 1/13 to 1/17 in order to set the textural properties and to highlight the possibility to control the texture of the 3D printed monoliths at nanoscale. After polymerization, drying and carbonization processes, strength, robust and consistent pure carbon monoliths were obtained (Figure 2(Aii)). During the carbonization process (900 °C/2 h/N_2_ flow), the template is removed, and the channels get open (Figure 2(Aiii)). The template was made using polyester, a thermofusible polymer. During carbonization, polyester is melted at around 190 °C and then, it decomposes at a range of 350–400 °C (see TGA, Appendix A) being removed from the channels. Note that CPE completely decomposed up to the carbonization temperature (900 °C) and no residue remains after the N_2_-treatment. Simultaneously, the xerogel loses oxygen and hydrogen functionalities and there is an enrichment in carbon, giving a highly pure carbon structure and the porosity is developed. After the carbonization process, a contraction of the monolith of ≈19 is observed, which is usual for RF carbon xerogels (≈23 without template). It is important to highlight that a perfect negative of the template is obtained without any breakage in the carbon walls (thickness of 0.6 mm) for both channels configurations, and the accuracy of the replica reaches the micrometer range (Figure 3A,B). This confirms the complete filling of the 3D-printed templates and the good wettability and affinity between the polymeric template and the polymerization mixture. Thus, this method allows the complete and perfect reproduction of the 3D printed design. It is important to note that we had success using polyester to print the molds, but several previous attempts performed with other polymers (like PLA filament) failed because the polymer material reacts with the carbon gel precursor mixture. 

To evaluate the scalability of the process, three times larger monoliths were successfully prepared, and the fidelity of the template reproduction is maintained. The integrity of the carbon monolith was preserved (Figure 2C), showing that full use of the larger-scale capabilities enabled by the combination of 3D-printing and sol-gel technologies is possible.

The porosity of carbon gel was controlled and fitted in the conventional design by modifying the sol-gel polymerization conditions, in this case by varying the resorcinol/water (R/W) ratio. The polycondensation of resorcinol-formaldehyde produces clusters of macromolecules that grow by reacting with each other to form primary colloidal particles. Initially, these colloidal particles are independent and are forming a suspension, which by Brownian motion begin to aggregate due to the reaction of their surface groups [24], forming an interlocking coral-like structure as shown in Figure 3C,D. Therefore, carbon gels are formed by interconnected primary particles, and this structure is responsible of the porous texture. Depending on the size and interconnection degree of these primary particles, the pore size distribution can be adjusted from micro, meso- to macro-pores. It can be seen that the growth rate becomes larger by decreasing R/W ratio, that is, the primary particles size increase and the interconnection between them decreases by decreasing the R/W ratio (Figure 3C,D), that is, increasing the amount of water. At low water concentration (R/W = 1/13, Figure 3C), primary particles reach better contact to each other and thus, a high-interconnected network of small particles is obtained creating the smallest size pores of the series (1000 nm). Conversely, at high water concentration (R/W = 1/17, Figure 3E), the interconnection is lower and the growth of particles higher, creating a less-interconnected network of bigger particles leaving larger pores (10,000 nm) as was also pointed out by Hg porosimetry (Figure 3F). This corroborates that the porosity of the carbon gel can be set by modifying the sol-gel polymerization conditions depending on the application necessities. Note that the size of the pores obtained lay in the macroporous range (>50 nm), and these large pores are very appropriate for further dispersion of the active phase particles, Ni/CeO_2_ in our case. To analyze the microporosity and narrow mesoporosity CO_2_ and N_2_ adsorption isotherms were also performed. The total micropores volume determined by N_2_ (0.01 cm^3^/g) is much lower that of the volume determined by CO_2_ (0.33 cm^3^/g) for all monoliths, which indicates diffusion restrictions to the ultramicroporosity (<0.7 nm). Moreover, as expected, these volumes are independent of the R/W ratio and the channels architecture (S_mic_ was ≈ 850 m^2^/g for all monoliths). The average micropores size is 0.65 nm for all monoliths. Therefore, the aqueous suspension of active phase cannot enter this porosity range and this ultramicroporosity has not a significant effect on the catalytic performance. 

Once the feasibility and possibilities of the proposed synthesis method have been proven, the advantages that these new and advanced materials can provide in catalysis have been tested. As a proof of concept, the carbon monoliths with conventional and advanced designs were impregnated with a water suspension of Ni/CeO_2_ active phase and were tested in the CO_2_ methanation reaction. This reaction was selected taking into account the interest in CO_2_ emissions reduction, since this reaction has been proposed as a feasible route to achieve a net zero carbon dioxide balance if both the hydrogen and energy involved are obtained by renewable sources. 

The same amount of active phase was deposited in all monoliths (115 ± 8 mg) for comparative reasons. The design does not affect the active phase dispersion; similar active phase dispersion was obtained for both designs with the same R/W ratio (Figure 4C,D). However, the dispersion of the Ni/CeO_2_ active phase on the carbon monoliths is strongly affected by the carbon porous texture (Figure 4 and Figure 5). Overall, the Ni/CeO_2_ dispersion increases as the R/W molar ratio decreases, that is, with the increase of the width and volume of macropores. Note that the Ni/CeO_2_ active phase is distributed more homogeneously on the channels and the walls of the monolith prepared with a R/W molar ratio of 1/17 (Figure 4C), as the diffusion of the active phase ethanolic suspension is favored by the macroporous structure. The opposite case occurs for the R/W molar ratio of 1/13 (Figure 4A), where the narrow macroporosity hinders the diffusion of the active phase inside the carbon network and active phase is mainly distributed on the channel surface forming agglomerated accumulations (Figure 4A center). An intermediate situation is observed for R/W molar ratio of 1/15. Some agglomerated accumulations can be observed on the channel surface (Figure 4B center), but the active phase is also distributed inside the carbon network (inside the monolith wall), but to a lesser extent than for molar ratio of 1/17. This different dispersion, and therefore, active phase-carbon contact, affects the reducibility of the active phase as it was observed by H_2_-TPR experiments (Appendix A). In the powdered catalyst, peaks observed in the temperature range of 300 to 450 °C can be attributed to the reduction of NiO into metallic nickel and partial reduction of de surface ceria. The displacement of the reduction bands to lower temperatures for the supported active phase indicated that the interaction of the Ni/CeO_2_ particles with the carbon favors the reducibility of the nickel and cerium species. This displacement is higher at lower R/W ratios manifesting that a greater dispersion of the Ni/CeO_2_ active phase increases the available active sites and their interaction with the carbon support and thus, enhances the active phase reducibility. It is also important to highlight that no significant changes in the chemical nature of the active phase was observed by Raman and XRD (Appendix A). This manifests that the deposition process does not affect the chemical nature of the Ni/CeO_2_ active phase.

A catalytic test with the carbon monolith without active phase was performed and no activity was observed. For Ni/CeO_2_ supported monolith, the porous texture and channels geometry of the monolithic supports [25] are crucial factors determining the catalytic performance (Figure 6A,B). On the one hand, the porous texture affects the active phase distribution, and thus the active phase-gas interaction and the active phase reducibility. Better dispersions were obtained by decreasing the R/W ratio due to the porosity broadening, and thus, CO_2_ conversion increases (Figure 6A). The best catalytic performance is obtained with the catalyst with R/W ratio 1/17, due to the best active phase dispersion and reducibility. On the other hand, the channels configuration also has a deep influence on the catalytic performance (Figure 6B). Conventional straight channels force laminar flow of the gas, creating radial diffusion limitations. However, the fluid encounters more intricate paths as long as it passes through the novel tortuous design, which favors the radial diffusion, and thus, the active phase-gas interaction. Consequently, the reaction rate obtained with the monolith with novel tortuous channels design is increased regarding the conventional honeycomb configuration with straight and parallel channels. Finally, an isothermal test was also performed at 300 °C to analyze the stability of the prepared carbon-based catalysts, observing that high activity (≈80%) and selectivity of methane (≈100) is maintained after several reaction hours (Figure 6C), denoting high stability.

These catalytic experiments demonstrate that the porosity and channel geometry control provided by this advanced technology can entail advantages in the catalytic performance of the active phases, and this concept can be extended to other reactions. This synthesis method is a very powerful tool for the design of high-performance catalyst supports that optimize an efficient use of usually expensive active phases.

Overall, our synthesis approach allows the manufacture of pure and integral carbon monoliths with specifically and rigorously designed geometry and porous texture, which improves the catalytic performance of conventional honeycomb monoliths. Although we have demonstrated our strategy for two type of channels architectures, complex-shaped morphologies can be obtained, which cannot be possible with other technologies. Therefore, a powerful tool is presented in this manuscript, which opens the gate to the manufacture of specifically designed materials based on each application demands, and a new world of possibilities in the design of materials is opened for a large field of catalytic reactions.

## Figures and Tables

**Figure 1 materials-14-05017-f001:**
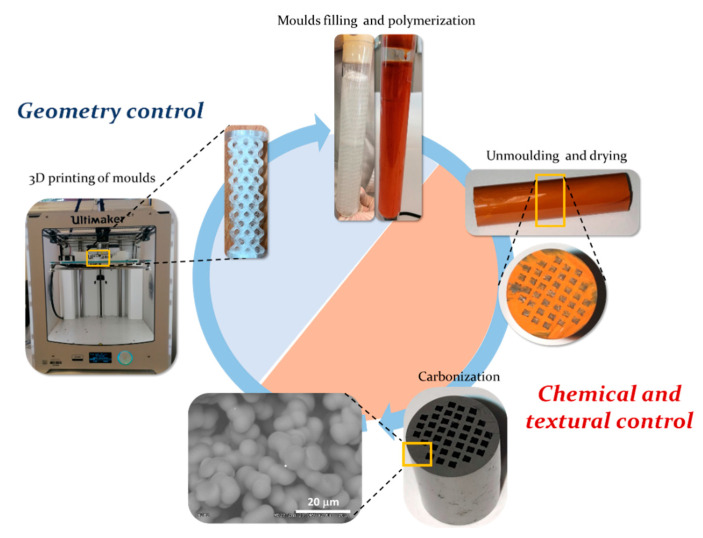
Scheme of the manufacture of 3D printed integral carbon monoliths with controlled geometry and chemical and textural properties.

**Figure 2 materials-14-05017-f002:**
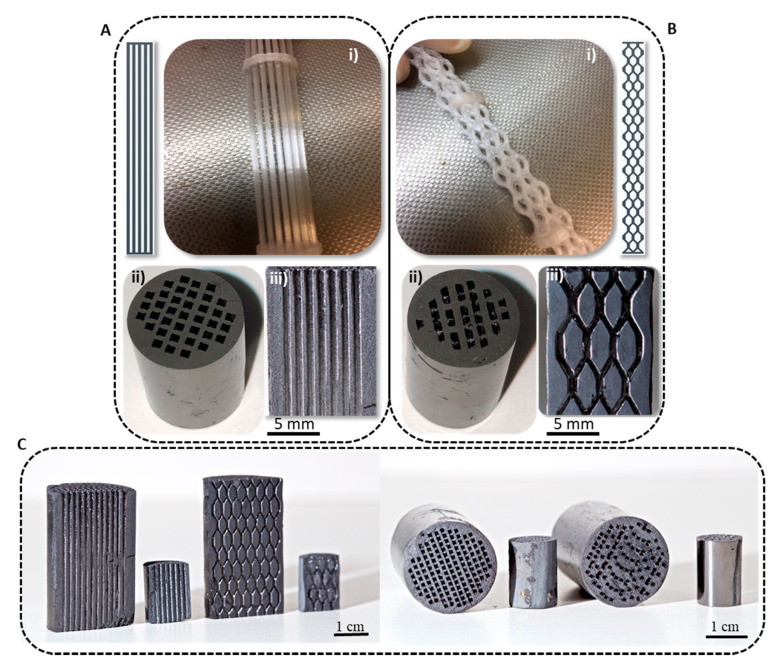
Integral carbon monoliths obtained by 3D printing with two different channels configurations (R/W = 1/17): Straight (**A**) and tortuous (**B**) channels. (**i**) 3D-printed template. (**ii**) Top view of the monoliths. (**iii**) Cross section of the monoliths. (**C**) Size scalability of carbon monoliths prepared using 3D printing.

**Figure 3 materials-14-05017-f003:**
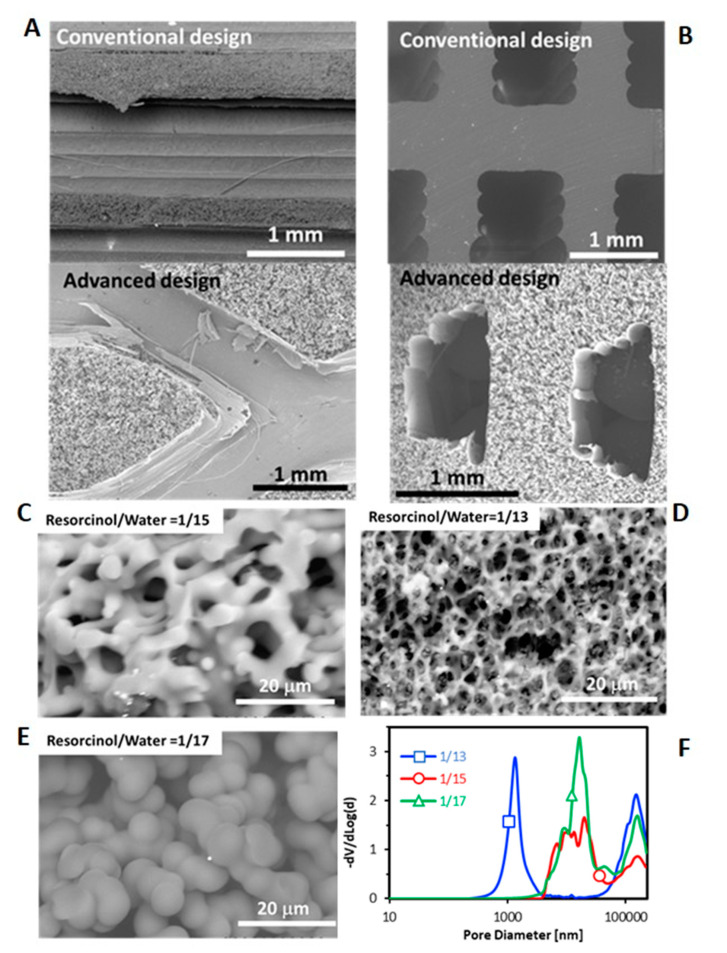
SEM images of 3D-printed monoliths with both channel configurations prepared with R/W molar ratio of 1/17 (**A**,**B**) and different R/W ratio for conventional design (**C**–**E**) and Hg-porosimetry plot (**F**). (**A**) longitudinal section and (**B**) Cross section.

**Figure 4 materials-14-05017-f004:**
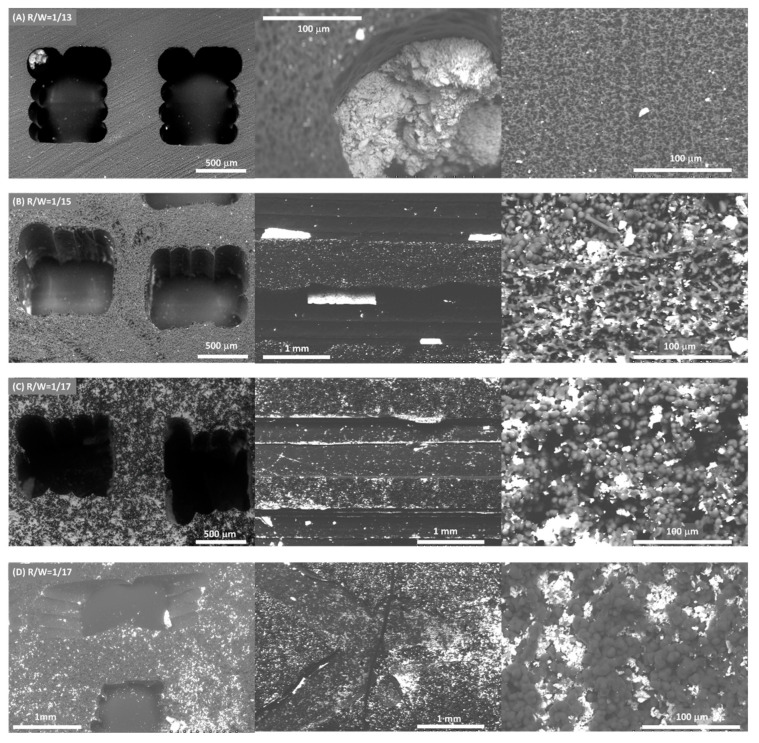
Distribution of Ni/CeO_2_ active phase (white) on the surface of carbon (grey) monoliths prepared with different R/W molar ratio for conventional design (**A**–**C**) and R/W = 17 for tortuous design (**D**). **Left**: Monolith transversal section, center: Monolith longitudinal section and **Right**: Inside channel wall.

**Figure 5 materials-14-05017-f005:**
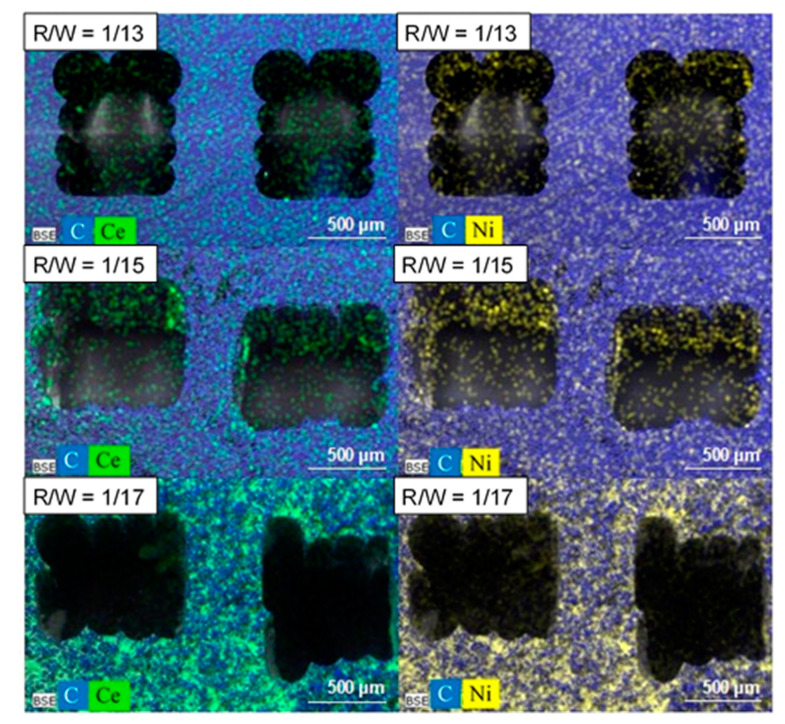
Chemical mapping of the Ni/CeO_2_ catalyst deposited on conventional carbon monoliths prepared with different R/W molar ratio. **Left**: C (blue) and Ce (green) and **Right**: C (blue) and Ni (yellow) elements.

**Figure 6 materials-14-05017-f006:**
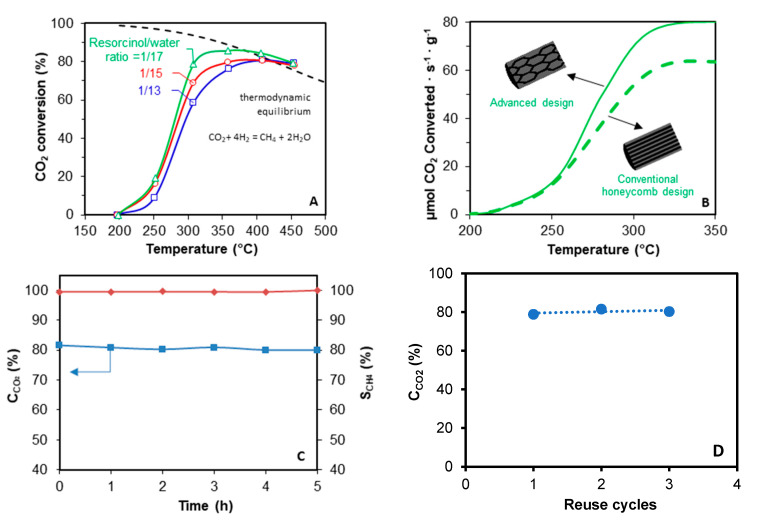
CO_2_ methanation tests over Ni/CeO_2_/carbon monolith catalysts prepared using 3D printing. (**A**) Influence of the porous texture of monoliths with conventional design on the catalytic performance. (**B**) Influence of the channels configuration on the CO_2_ methanation reaction rate; both monoliths were prepared with resorcinol/water ratio = 1/17. (**C**) Isothermal test at 300 °C for the catalyst with advanced design prepared with resorcinol/water ratio =1/17. C_CO2_ = CO_2_ conversion and S_CH4_ = Selectivity to methane. (**D**) Study of the stability of the catalysts after being reused in successive cycles. C_CO2_: conversion of CO_2_ at 300 °C.

## Data Availability

The data presented in this study are available on request from the corresponding author.

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
