# Peer review of "High Performance Tunable Catalysts Prepared by Using 3D Printing"

_materials, 2021, doi:10.3390/ma14175017_

Round 1

Reviewer 1 Report

This is a nice exploration on 3D printing of structured catalysts , and the support has unique structure, the authors coated Ni/CeO2 over the printed structure catalyst and tested for CO2 to methane. The printed catalyst clearly has better CO2 conversion activity, and the filling formulation has some effect on the performance. It is a quite novel work and can be accepted for publication after the following are considered:

  1. Can more details be provided for the catalyst test? Is the structured catalyst grounded and sieved or tested as structured catalyst? How much was used and what is the space velocity?
  2. Is the substrate of printed and extrudated supports the same? Does the substrate have effect on the catalyst performance?
  3. Can the image of the coated Ni/CeO2 be provided over the different substrate?
  4. Fig 4-B is a bit difficult understand for the conventional honey comb designed catalyst, which has lower CO2 converted rate at higher temperature, how can this be explained,
  5. How about catalyst coating after catalyst test ? are there any changes occurred to the coating?

In summary, it is a nice work, but more data might be better for quality improvement.

Author Response

This is a nice exploration on 3D printing of structured catalysts , and the support has unique structure, the authors coated Ni/CeO2 over the printed structure catalyst and tested for CO2 to methane. The printed catalyst clearly has better CO2 conversion activity, and the filling formulation has some effect on the performance. It is a quite novel work and can be accepted for publication after the following are considered:

Firstly, we sincerely are thankful for the review of this manuscript as well as for all your comments and suggestions. All changes in the revised manuscript have been highlighted by giving the text a yellow background.

  1. Can more details be provided for the catalyst test? Is the structured catalyst grounded and sieved or tested as structured catalyst? How much was used and what is the space velocity?

More details about the catalytic test have been provided in the revised manuscript. The structured catalysts were used as a monolith. The monoliths have a dimension of 1 cm of diameter x 1.5 cm of length (0.650 g). The amount of active phase (Ni/CeO2) incorporated and used in both monoliths was 110 ± 5 mg (0.17 g active phase/g of monolith); total weight of supported catalyst was 0.760 g. The gas hourly space velocity (GHSV) was 5093 h-1.

2. Is the substrate of printed and extrudated supports the same? Does the substrate have effect on the catalyst performance?

The carbon monolithic supports were prepared by polymerization of resorcinol and formaldehyde in presence of 3D printed polymeric templates which provide the channels architecture. This polymeric template is removed during the carbonization of the organic gel-based monolith. A catalytic test with the carbon monolith without active phase was performed and no activity was observed.

3. Can the image of the coated Ni/CeO2 be provided over the different substrate?

SEM images of Ni/CeO2 supported on the different monoliths have been included in and discussed in the revised manuscript (new Figures 4 and 5).

4. Fig 4-B is a bit difficult understand for the conventional honey comb designed catalyst, which has lower CO2 converted rate at higher temperature, how can this be explained,

In this figure, both designs are compared at same condition of synthesis (R/W=1/17) and active phase dispersion in order to properly analyze the influence of the channels configuration on the CO2 methanation reaction rate. Therefore, in this figure, it can be seen that the reaction rate obtained with the monolith with novel tortuous channels design is increased regarding the conventional honeycomb configuration with straight and parallel channels.In order to avoid misunderstanding, the experimental section was rewritten as well as all Figure captions.

Figure 6. (B) Influence of the channels configuration on the CO2 methanation reaction rate; both monoliths were prepared with resorcinol/water ratio =1/17.

5. How about catalyst coating after catalyst test ? are there any changes occurred to the coating?

Several recycling tests were performed with the same monolith (advanced monolith) and no differences were observed in the catalytic performance. Consequently, we can conclude that changes were not occurred after the catalytic test.

Figure 6. (D) Study of the stability of the catalysts after being reused in successive cycles. CCO2: conversion of CO2 at 300 ºC.

In summary, it is a nice work, but more data might be better for quality improvement.

Thank you very much again for your revision and comments

Reviewer 2 Report

This manuscript developed carbon integral monoliths with a complex network 16 of interconnected channels via 3D printing. The catalyst have been tested for CO2 methanation after loading a Ni/CeO2, and the activity enhanced by 25% at 300 ºC because the turbulent flow into the channels network. This manuscript results and discussions are lack of detailed explanations, and this research is absent of innovation. Unfortunately, I cannot recommend the manuscript publication in materials. Follow are some suggestions:

  1. The Ni/CeO2 catalyst most used for CO2 hydrogenation to methane, the 3D carbon integral monoliths used as support for catalyst, but CeO2 materials were synthesized in many morphologies in several researches, including the 3D structures. Therefore, this manuscript is absent of innovation.
  2. The author prepared series ratio of resorcinol/water molar, 1/13, 1/15 and 1/17, the activity also enhanced with the ratio increased. However, the resorcinol/water cannot as active sites or participated in this reaction, the obvious distinction is the pore size among these catalysts, and this may absent of sufficient evidences and discussions.
  3. Fig.3 C-E shown SEM images of 3D-printed monoliths, the SEM results cannot agree with the Fig.3F, 1/13 and 1/15 catalysts have bigger pore size than1/17.
  4. Fig.4 shown the conversion of CO2, the activity is higher than thermodynamic equilibrium, this may unreasonable.

Author Response

This manuscript developed carbon integral monoliths with a complex network 16 of interconnected channels via 3D printing. The catalyst have been tested for CO2 methanation after loading a Ni/CeO2, and the activity enhanced by 25% at 300 ºC because the turbulent flow into the channels network. This manuscript results and discussions are lack of detailed explanations, and this research is absent of innovation. Unfortunately, I cannot recommend the manuscript publication in materials. Follow are some suggestions:

Firstly, we sincerely are thankful for the review of this manuscript as well as for all your comments and suggestions. All changes in the revised manuscript have been highlighted by giving the text a yellow background.

1. The Ni/CeO2 catalyst most used for CO2 hydrogenation to methane, the 3D carbon integral monoliths used as support for catalyst, but CeO2 materials were synthesized in many morphologies in several researches, including the 3D structures. Therefore, this manuscript is absent of innovation.

We agree with the reviewer, many researchers, including our research group (Applied Materials Today 19 (2020) 100591; Fuel Processing Tech 2012 (2021) 106637), are focused on the development of active phases with different morphologies, including 3D structures such as 3DOM, nanoparticles, nanocubes, etc for CO2 methanation. However, catalysts are always tested in powder form. Nonetheless, for a real industrial application of heterogeneous catalysts honeycomb monoliths is the preferred format due to kinetic, fluid-dynamic and diffusional reasons. However, the geometry of these monoliths is limited to designs obtained by extrusion, generally straight channels that impose a laminar flow regime thorough them and, therefore, less efficiency of the usually expensive active phases. The use of the 3D-printing technology to the design of monoliths with specific configurations would be a breakthrough in the field of catalysis but it is rarely studied. Therefore, in the present work we are not focused on the synthesis and improvement of the methanation active phase. Our aim is to obtain advanced non-straight channeled monolithic supports for the CO2 methanation active phases that cannot be obtained by traditional methods like extrusion using 3D printing and to analyze the effect of the channel morphology and textural properties of these novel carbon-based monolithic supports in the CO2 methanation. Thus, in this case, we select a conventional Ni/CeO2 active phase. Therefore, the impact and innovation of the manuscript is ensured.

2. The author prepared series ratio of resorcinol/water molar, 1/13, 1/15 and 1/17, the activity also enhanced with the ratio increased. However, the resorcinol/water cannot as active sites or participated in this reaction, the obvious distinction is the pore size among these catalysts, and this may absent of sufficient evidences and discussions.

Obviously, Resorcinol/water cannot act as active sites or participate in the CO2 methanation since there are only reagents for the synthesis of the catalytic support: carbon gel monoliths. However, the modification of the synthesis conditions of organic xerogels (then transformed into carbon xerogels by carbonization), in this case the R/W ratio, strongly affects the textural properties of the carbon monolith and consequently, the active phase (Ni/CeO2) dispersion along the channels and walls of the carbon monolith (see SEM and mapping images). This better dispersion or distribution of active phases on the carbon support surface, affects the gas/active phase contact and therefore, the catalytic activity.

3. Fig.3 C-E shown SEM images of 3D-printed monoliths, the SEM results cannot agree with the Fig.3F, 1/13 and 1/15 catalysts have bigger pore size than1/17.

New SEM images have been included in Figure 4 and 5, where it can be observed that the wall porosity increased and become wider by decreasing the R/W ratio, this is, by increasing the amount of water (13, 15 and 17)

4. Fig.4 shown the conversion of CO2, the activity is higher than thermodynamic equilibrium, this may unreasonable.

As it can be observed in the Figure, the last point (at high temperature) is slightly higher that the marked by the thermodynamic equilibrium. This is within the experimental error. The concentrations must be corrected because flow changes during the reaction as a result of the water condensation for that, this error can be higher at high temperature in which high production of water occurs.

Round 2

Reviewer 2 Report

The authore revised the manuscript following some of the previous suggestions, However, the important points they did nothing to improve.

However, I can not agree with their response to question 1 and 4.

Concerning the CO2 methanation with Ni and Ni/CeO2, the related articles are not fully cited. Journal of COâ‚‚ Utilization 34 (2019) 676–687, and  https://doi.org/10.1021/acs.iecr.1c01953

 In addition, the conversion of CO2, the activity is higher than thermodynamic equilibrium, this can not be explained by experimental error..

Author Response

The authors response is attached in a word file

Round 3

Reviewer 2 Report

The authors have revised the manuscript following all my suggestions, I agree for its publication in the present form.